# Spectral Co-Distillation for Personalized Federated Learning

**Zihan Chen[1], Howard H. Yang[2], Tony Q.S. Quek[1], and Kai Fong Ernest Chong[1] ***

[1]Singapore University of Technology and Design (SUTD)

[2]Zhejiang University/University of Illinois Urbana-Champaign Institute, Zhejiang University

`zihan_chen@sutd.edu.sg`

## Abstract

Personalized federated learning (PFL) has been widely investigated to address the challenge of data heterogeneity, especially when a single generic model is inadequate in satisfying the diverse performance requirements of local clients simultaneously. Existing PFL methods are inherently based on the idea that the relations between the generic global and personalized local models are captured by the similarity of model weights. Such a similarity is primarily based on either partitioning the model architecture into generic versus personalized components, or modeling client relationships via model weights. To better capture similar (yet distinct) generic versus personalized model representations, we propose *spectral distillation*, a novel distillation method based on model spectrum information. Building upon spectral distillation, we also introduce a co-distillation framework that establishes a two-way bridge between generic and personalized model training. Moreover, to utilize the local idle time in conventional PFL, we propose a wait-free local training protocol. Through extensive experiments on multiple datasets over diverse heterogeneous data settings, we demonstrate the outperformance and efficacy of our proposed spectral co-distillation method, as well as our wait-free training protocol.

## 1 Introduction

With the rapid rise in mainstream popularity of artificial intelligence (AI) models such as ChatGPT [1] and LoRA [2], there has been an increasing shift towards the development of personalized AI assistants [3]. Hence, in a future where personalized AI services become mainstream, training AI models on personal data while preserving data privacy would become increasingly important [4], and maintaining the quality for such models would require collaborative training across multiple models. Personalized federated learning (PFL) emerges as a promising privacy-preserving distributed learning paradigm that is well-equipped to meet such requirements [3]. As an extension of federated learning (FL), PFL aims to train a customized machine learning model for each client or each group of clients with similar preferences [5]. When faced with inconsistencies in the objective functions of different clients, conventional FL fails to generalize well with just a single model, while in contrast PFL promises to generalize well across all clients, even in the presence of data heterogeneity (e.g., label distribution skew and label quantity skew) [6–14].

To tackle the challenges of personalization, numerous works have focused on designing new PFL systems, or enhancing the performance of personalized models from different aspects, such robustness, fairness, and model convergence[15–18]. Under federated settings, personalization is achieved through capturing the (dis-)similarity of the local versus globally shared model representations. In

---

*Corresponding author

37th Conference on Neural Information Processing Systems (NeurIPS 2023).

practical FL/PFL applications of collaboratively training deep neural networks (DNNs), only the model parameters (e.g., model weights or gradients) are exchanged between the clients and the server [8, 19]. Existing DNN-based PFL methods capture this (dis-)similarity either by decoupling the model architecture into groups of layers/channels[20–25], or by designing local optimization methods with regularization based directly on model weights [17, 15]. Unfortunately, the motivations for such approaches are based on empirical observations, without an overarching theory to explain model (dis-)similarity in relation to training dynamics.

In deep learning theory, the training dynamics of DNNs have been studied from the lens of Fourier analysis [26]. A crucial insight from this analysis is that there is an implicit self-regularization effect arising from the training process itself. Given a target function $f$ to learn, the model tends to learn the lower frequencies of the Fourier spectrum of $f$ first before learning the respective higher frequencies. Such a bias in this training process is called *spectral bias* [27, 28]. Informally, spectral bias describes the commonly encountered phenomenon that DNNs first learn low-level features before learning high-level features.

Motivated by this insight, we can distinguish different levels of features in a model representation by looking at its Fourier spectrum. Intuitively, diverse personalized models would still share the same low-level features, and a global generic model would contain the same low-level features. Hence, despite any inconsistencies in the objective functions of different clients, there would be no conflict in learning low-level features for both the generic and personalized models. Consequently, with the expected similarity in the lower frequency components of the Fourier spectra of both the generic and personalized models, we can distill the knowledge of the lower Fourier coefficients to boost the performance of the generic model. Dually, the entire Fourier spectrum of the generic model, which includes the "averaged" high-level features across all clients, would benefit the training of the personalized models. By combining both perspectives, *we shall propose a co-distillation framework for PFL that captures (dis-)similarity in models via spectral information.*

Typically, when designing PFL systems, a compute-and-wait protocol is implicitly assumed for local training [15, 23]. This means that the locally updated generic models would be sent by the clients to the server after all local computation tasks have been completed. Such a protocol would yield a period of idle waiting where clients have to wait for the next aggregated model to be broadcasted. *By circumventing this compute-and-wait protocol, we shall utilize the local idle time for training to reduce the total PFL runtime.*

Overall, our contributions can be summarized as follows:

- We propose a spectral co-distillation framework for PFL. In particular, this is the first ever use of spectral distillation in PFL to capture the (dis-)similarity of the generic and personalized models. Also, this is the first ever bi-directional knowledge distillation directly between the generic and personalized models.
- We propose a wait-free local training protocol for our spectral co-distillation framework, where we utilize the idle time during global communication so as to reduce the total PFL runtime.
- Through extensive experiments on multiple datasets with heterogeneous data settings, we demonstrate the outperformance and efficacy of our proposed spectral co-distillation framework with the wait-free communication protocol design for PFL, with respect to model generalizability and the total PFL runtime.

## 2 Related work

**PFL.** In PFL, prior efforts have focused on training multiple personalized models via leveraging the similarity and relationships between the global generic model and the local personalized models, such as via model interpolation/mixture [29], model decoupling [22], and personalized optimization with customized regularizers [15]. In DNN-based FL applications, decoupling-based approaches divide the model into a private part (kept at the local side) and a shared part (exchanged between the server and clients) [3, 25, 23]. In particular, FedPer [22] and FedRep [24] share the shallow layers and train personalized deep layers, while in contrast, LG-Fed [20] and $CD^2$-pFed maintain personalized shallow layers and channels [21], respectively. Moreover, Fed-RoD proposes a framework to achieve state-of-the-art (SOTA) performance for generic and personalized models simultaneously, based on the "two-loss, two-predictor" design[23]. APFL [5] and L2GD [29] consider using a mixture of local

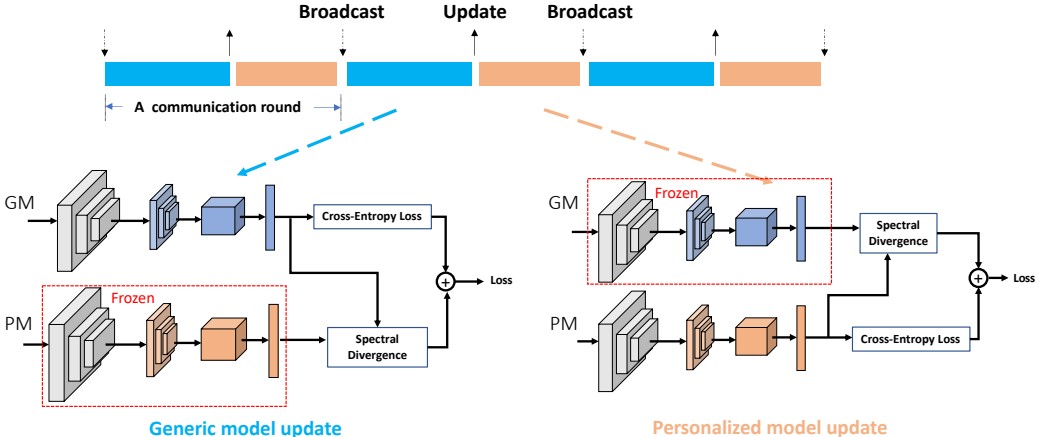

Figure 1: Spectral co-distillation framework with wait-free local training for PFL, in which the generic model (GM) training and the personalized model (PM) training are carried out via spectral distillation in two different stages.

and global models to achieve personalization, in which the mixture weight controls the personalization level. Personalized local training methods have been recently explored, which include local fine-tuning in FedBABU [30], bi-level optimization in Ditto [15], feature alignment in FedPAC [31], and personalized model sparsification in FedMask [16, 32] and PerFedMask [33]. More broadly, meta-learning [34, 35], gaussian processes [36], and hyper-network-based approaches [37] have been investigated in PFL. Specifically, there is another type of PFL that aims to train personalized models at the level of clusters of clients with similar preferences [38–40].

**Knowledge Distillation (KD) in FL.** KD has been widely explored in knowledge transfer scenarios, which usually is used to transfer knowledge from the pre-trained teacher model to the student model via minimizing the distance from the latent or logit outputs of the two models [41, 42]. KD-based FL frameworks have been developed with diverse setups, such as FedMD [43] and FedDF [44]. On the other hand, knowledge-transfer-based PFL frameworks are investigated in [45, 46] with different model structures at the local clients, which could address the system heterogeneity and improve communication efficiency. However, such methods rely on the assumption of having access to a public labeled/unlabelled dataset, which may not be a realistic assumption in FL applications [3]. Moreover, co-distillation methods have been investigated in communication-efficient decentralized scenarios to improve generalizability [45].

## 3 Proposed framework

The main goal of this work is to train a generic global model and multiple personalized models simultaneously. As summarized in Sec. 1, our proposed framework consists of three major components: spectral distillation-based personalized model training, spectral co-distillation-based generic model training, and the wait-free sequential computation-communication protocol. In this section, we first provide the preliminary and problem formulation for PFL and model spectrum in Sec. 3.1. Next, we present our proposed spectral distillation approach for PFL in Sec. 3.2, co-distillation-based generic model training in Sec. 3.3, and the wait-free local training protocol in Sec. 3.4, accordingly. Moreover, the summarized algorithm is given in Sec. 3.5.

### 3.1 Preliminaries

**Problem formulation for FL and PFL.** Consider an FL system consisting of a server and $N$ clients, in which client $i$ has a loss function $f_i : \mathbb{R}^d \to \mathbb{R}$ used for training on its local private dataset $\mathcal{D}_i = \{(x_i^j, y_i^j)\}_{j=1}^{n_i}$, where $n_i = |\mathcal{D}_i|$ denotes the size of the local dataset of client $i$. In conventional FL, the objective of all the participating clients in this system is to find a global model $w \in \mathbb{R}^d$ that

solves the following minimization problem [19]:

$$\underset{w \in \mathbb{R}^d}{\text{minimize}} \left\{ F(w) := \sum_{i=1}^{N} \frac{n_i}{n} f_i(w) \right\}, \tag{1}$$

where $n = \sum_{i=1}^{N} n_i$ is the total number of training samples across the $N$ clients. In a typical communication round $t$, a subset $\mathcal{S}_t$ of clients is selected to conduct local training, starting from the latest global model weights $w_G^t$. Let $w_i^t$ denote the weights of client $i$'s model after local training. At the end of communication round $t$, the server would collect local models from the selected clients to update the global model via Federated Averaging (FedAvg), i.e. $w_G^{t+1} \leftarrow \sum_{i \in \mathcal{S}_t} p_i^t w_i^t$, in which $p_i^t = n_i / \sum_{k \in \mathcal{S}_t} n_k$ represents the ratio of the local data samples in client $i$ over the total number of data samples in the selected subset $\mathcal{S}_t$ of clients for communication round $t$.

There are two general types of PFL: a) training $N$ personalized models for all $N$ clients; and b) training 1 generic model and $N$ personalized models simultaneously. In this work, we investigate the latter one, which we term as "PFL+". This means each client $i$ has a local personalized model $w_{p,i}$ for its private dataset $\mathcal{D}_i$, and all clients jointly participate in the training of the generic model $w_G$. After local training at client $i$, the updated generic model is denoted by $w_{G,i}$. Thus, PFL can be formulated using a regularized loss function with regularization term $R_p(w_{p,i}, w_{G,i})$. For example, $R_p(w_{p,i}, w_{G,i})$ could represent the similarity/divergence between the global and local models' features, such as model weights, feature centroids, and prototypes. In our method, $R_p(w_{p,i}, w_{G,i})$ represents cross-model distillation during the training of client $i$'s personalized model. Therefore, the objective of personalized model training in PFL+ can be formally formulated as a bi-level optimization problem [17]:

$$(\textbf{P1}): \quad \underset{w_{p,i} \in \mathbb{R}^d}{\text{minimize}} \quad \left\{ f_{p,i}(w_{p,i}) := f_i(w_{p,i}) + \lambda_p R_p(w_{p,i}, w_{G,i}) \right\} \quad \text{for each client } i \tag{2}$$

$$\text{subject to} \quad w_{G,i} \leftarrow \text{ updated generic model from } w_G, \tag{3}$$

where the regularization coefficient $\lambda_p$ is used to control the level of personalization. For client $i$, when referring to a specific communication round $t$, we shall denote the untrained personalized model and updated generic model by $w_{p,i}^{t-1}$ and $w_{G,i}^t$, respectively.

## 3.2 Personalized local model training

Motivated by both theoretical and empirical insights of the spectral bias inherent in the training dynamics of DNNs, we explore the use of the Fourier spectrum of the generic model for knowledge distillation to enhance the training of personalized local models. In particular, we propose a distillation regularization term representing the divergence between the *full* model spectra of the generic and personalized models.

First, we introduce some notation. Given vectors $p = (p_1, \ldots, p_d)$, $q = (q_1, \ldots, q_d)$ in $\mathbb{R}^d$, define the divergence function $\mathfrak{D}(p \| q) := \sum_{i=1}^{d} p_i \log p_i - p_i \log q_i$. (By convention, $0 \log 0 := 0$.) Note that when $p$ and $q$ are stochastic vectors representing parameter vectors of multinomial distributions $P$ and $Q$, then $\mathfrak{D}(p \| q)$ is identically the Kullback–Leibler (KL) divergence from $P$ to $Q$. Next, let $\texttt{DFT} : \mathbb{C}^d \to \mathbb{C}^d$ denote discrete Fourier transform, let $\varrho : \mathbb{C}^d \to \mathbb{R}^d$ be the map given by $(z_1, \ldots, z_d) \mapsto (\|z_1\|, \ldots, \|z_d\|)$, and define the function $s : \mathbb{R}^d \to \mathbb{R}^d$ by $s := \varrho \circ \texttt{DFT}$. For an input vector of the weights of a DNN model, the output vector after applying $s$ shall be called the *spectrum vector* of that model [28]. Thus, in communication round $t$, the spectrum vectors of the personalized model $w_{p,i}^{t-1}$ of client $i$ and updated generic model $w_{G,i}^t$ are written as $s(w_G^t)$ and $s(w_{p,i}^{t-1})$, respectively. We shall represent the divergence of the personalized and generic models by $\mathfrak{D}(s(w_{p,i}^{t-1}) \| s(w_{G,i}^t))$, the divergence of their spectrum vectors.

Concretely, we define $R_p(w_{p,i}, w_{G,i}) := \mathfrak{D}(s(w_{p,i}^{t-1}) \| s(w_{G,i}^t))$, and let $f_i$ be the cross-entropy loss $\mathcal{L}_{CE}$ for all $i$. Then the personalized objective function $f_{p,i}$ of client $i$ in communication round $t$ (cf. (2)) is given by:

$$\mathcal{L}^p(w_{p,i}^{t-1} | w_{G,i}^t) := \mathcal{L}_{CE}(w_{p,i}^{t-1} | \mathcal{D}_i) + \lambda_p \mathfrak{D}(s(w_{p,i}^{t-1}) \| s(w_{G,i}^t)). \tag{4}$$

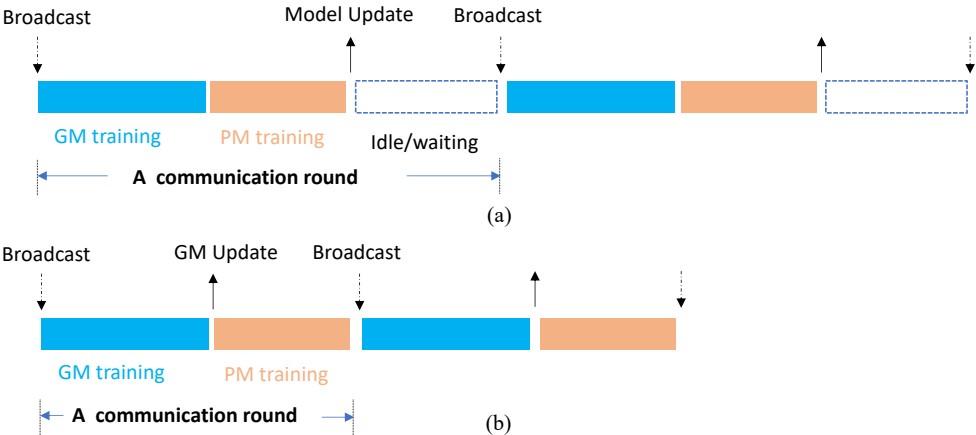

Figure 2: A comparison of the (a) conventional compute-and-wait protocol with the (b) proposed wait-free training protocol.

For simplicity, we use a common time-invariant $\lambda_{\mathrm{p}}$ for all clients throughout training. Since we are distilling the knowledge of the spectrum vector $s(w_{\mathrm{G},i}^t)$ in (4), we term our approach as *spectral distillation*.

### 3.3 Generic model training

Given a PFL+ training framework, it is natural to connect the *roles of generic and personalized models* to the *roles of the teacher and student models in distillation*, where the training of one model is guided by the knowledge distilled by the other. Co-distillation extends this idea. Intuitively, the role of each model alternates between teacher and student for knowledge distillation during training. In PFL+, since we are concurrently training both the generic and personalized models, either of them could be used for knowledge distillation. The key challenge for applying co-distillation to PFL+ is that it is not obvious what knowledge should be distilled from the personalized models to enhance the training performance of the generic model.

In the theory of deep learning, it is well-known that when training a DNN, there is a learning bias towards the lower frequencies of its Fourier spectrum [27, 28]. In fact, the lower-frequency components of this spectrum are robust to random weight perturbations. Hence, with diverse personalized models, we would still expect the lower-frequency components of the spectra of all models (both generic and personalized) to be similar. Consequently, we could use such lower-frequency components for knowledge distillation to enhance generic model training.

Motivated by this, we propose a truncated spectrum-based distillation loss as the regularizer for generic model training. Given $0 < \tau \leq 1$, let $\imath_\tau : \mathbb{R}^d \to \mathbb{R}^{\lceil \tau d \rceil}$ be the projection map onto the first $\lceil \tau d \rceil$ entries, and define $\widehat{s} := \imath_\tau \circ s$. Then the loss function for generic model training, which depends on the truncated spectrum vectors $\widehat{s}(w_{\mathrm{G},i}^t)$ and $\widehat{s}(w_{\mathrm{p},i}^{t-1})$, is given by:

$$\mathcal{L}^{\mathrm{G}}(w_{\mathrm{G},i}^t | w_{\mathrm{p},i}^{t-1}) := \mathcal{L}_{\mathrm{CE}}(w_{\mathrm{G},i}^t | \mathcal{D}_i) + \lambda_{\mathrm{g}} \mathfrak{D}(\widehat{s}(w_{\mathrm{G},i}^t) \| \widehat{s}(w_{\mathrm{p},i}^{t-1})), \tag{5}$$

where the regularization term $R_{\mathrm{G}}(w_{\mathrm{G},i}^t, w_{\mathrm{p},i}^{t-1}) := \mathfrak{D}(\widehat{s}(w_{\mathrm{G},i}^t) \| \widehat{s}(w_{\mathrm{p},i}^{t-1}))$ depends on the hyperparameter $\tau$, and $\lambda_{\mathrm{g}}$ is the coefficient of this regularization term. Analogous to **(P1)**, the objective of generic model training in PFL+ could be formulated as the following bi-level optimization problem:

$$\textbf{(P2):} \quad \underset{w_{\mathrm{G}} \in \mathbb{R}^d}{\text{minimize}} \quad \left\{ f(w_{\mathrm{G}}) := \sum_{i=1}^{N} \frac{n_i}{n} \left( f_i(w_{\mathrm{G}}) + \lambda_{\mathrm{G}} R_{\mathrm{G}}(w_{\mathrm{G}}, w_{\mathrm{p},i}) \right) \right\} \tag{6}$$

$$\text{subject to} \quad w_{\mathrm{p},i} \leftarrow \text{output of } \textbf{(P1)} \text{ for client } i, \text{ for } i = 1, \dots, N. \tag{7}$$

Overall, by combining the two spectral distillation approaches together, we get a training framework for PFL+, which we shall call *spectral co-distillation*.

**Algorithm 1** Spectral Co-Distillation with Wait-free Training for PFL+

---

**Inputs:** $N, T, \eta_{\mathrm{p}}, \eta_{\mathrm{G}}, w_{\mathrm{G}}^0, \{w_{\mathrm{p},i}^0\}_{i=0}^N, E_{\mathrm{G}}, E_{\mathrm{p}}$
**Outputs:** Generic model $w_{\mathrm{G}}^T$, personalized models $\{w_{\mathrm{p},i}^T\}_{i=1}^N$

1: **for** $t = 1$ **to** $T$ **do**
2:     **for** each client $k = 1$ **to** $N$ **in parallel do**
       *// Generic model training and update*
3:        $w_{\mathrm{G},k}^t \leftarrow \mathrm{GMUPDATE}(w_{\mathrm{G}}^{t-1}, w_{\mathrm{p},k}^{t-1})$
4:        Upload weights $w_{\mathrm{G},k}^{(t)}$ to server
       *// Personalized model training* (Task 1: Line 5)
5:        $w_{\mathrm{p},k}^t \leftarrow \mathrm{PMUPDATE}(w_{\mathrm{p},k}^{t-1}, w_{\mathrm{G},k}^t)$
6:        **do sequentially** (Task 2: Lines 7–8)
7:           Uplink communication of generic model to server     Perform Tasks 1 & 2 in parallel
8:           Generic model aggregation to obtain $w_{\mathrm{G}}^t$ at server
9: **return** $w_{\mathrm{G}}^T, \{w_{\mathrm{p},i}^T\}_{i=1}^N$

**function** $\mathrm{GMUPDATE}(w_{\mathrm{G}}^{t-1}, w_{\mathrm{p},k}^{t-1})$
    **Require:** $w_{\mathrm{G}}^{t-1}, w_{\mathrm{p},k}^{t-1}$ are the latest generic model and personalized model.
1: $w_0 \leftarrow w_{\mathrm{G}}^{t-1}$
2: **for** $j = 1$ **to** $E_{\mathrm{G}}$ **do**
3:     $w_j \leftarrow w_{j-1} - \eta_{\mathrm{G}} \nabla \mathcal{L}^{\mathrm{G}}(w_{j-1} | w_{\mathrm{p},k}^{t-1})$     *// Using truncated low frequency spectrum information*
4: **return** $w_j$

**function** $\mathrm{PMUPDATE}(w_{\mathrm{p},k}^{t-1}, w_{\mathrm{G},k}^t)$
    **Require:** $w_{\mathrm{p},k}^{t-1}, w_{\mathrm{G},k}^t$ are the latest personalized model and updated generic model of client $k$.
1: $w_0 \leftarrow w_{\mathrm{p},k}^{t-1}$
2: **for** $j = 1$ **to** $E_{\mathrm{p}}$ **do**
3:     $w_j \leftarrow w_{j-1} - \eta_{\mathrm{p}} \nabla \mathcal{L}^{\mathrm{p}}(w_{j-1} | w_{\mathrm{G},k}^t)$     *// Using full model spectrum information*
4: **return** $w_j$

---

### 3.4   Wait-free Local Training Protocol

In the context of federated computing, the total runtime, which includes both local computation and communication time throughout the entire training process, is a direct indicator of communication efficiency. However, current PFL frameworks adopt a compute-and-wait protocol for local training. This means that in each round, the client performs both generic and personalized model updates only after all local computation tasks have been completed, and resumes local training upon receiving the latest global model broadcasted from the server. In consequence, there is idle waiting time between model update and model broadcast; see Fig. 2(a).

To improve the communication efficiency of PFL training with respect to the total runtime, we propose a *wait-free local training protocol*, as depicted in Fig. 2(b). In our protocol, the client updates the generic model according to the conventional generic FL training and trains the personalized model during the global communication time period. Unlike existing PFL frameworks, local clients would send the updated generic model to the server before the start of the personalized model training. Thus, our protocol eliminates idle waiting time, thereby dramatically reducing total runtime. Furthermore, it could be easily incorporated into existing PFL frameworks, such as Ditto [15], to further improve the efficiency; see Tab. 4.

**Discussion on the proposed protocol and related work.** Our proposed wait-free local training protocol is specially designed for the PFL+ scenario, where each client trains two models locally. For simplicity, we use this protocol in our experiments, under the assumption of synchronized PFL+. For comparison in the asynchronized PFL+ setting [47], see Appendix. Related work that reduce the total training runtime, such as delayed gradient averaging [48] and wait-free decentralized FL training [49], are designed for conventional FL and does not deal with the PFL+ scenario. Furthermore, we also provide a discussion on how our wait-free local training protocol could be adapted to the partial client participation scheme in FL in the Appendix.

| Methods | $\alpha = 1$ | | $\alpha = 0.5$ | | $\alpha = 0.1$ | |
|---------|------|------|------|------|------|------|
| | GM | PM | GM | PM | GM | PM |
| FedAvg | $85.35 \pm 0.11$ | $(80.33 \pm 0.38)$ | $80.76 \pm 0.13$ | $(74.51 \pm 0.48)$ | $73.51 \pm 0.17$ | $(72.68 \pm 0.39)$ |
| FedProx | $85.61 \pm 0.08$ | $(86.28 \pm 0.21)$ | $80.54 \pm 0.14$ | $(76.88 \pm 0.30)$ | $71.96 \pm 0.12$ | $(73.77 \pm 0.30)$ |
| FedDyn | $86.03 \pm 0.13$ | $(85.33 \pm 0.19)$ | $80.88 \pm 0.18$ | $(78.93 \pm 0.25)$ | $73.62 \pm 0.14$ | $(74.25 \pm 0.58)$ |
| FedGen | $86.17 \pm 0.32$ | $(\ 85.24 \pm 0.47)$ | $79.86 \pm 0.34$ | $(77.52 \pm 0.43)$ | $71.36 \pm 0.28$ | $(71.42 \pm 0.63)$ |
| FedAvgM | $85.44 \pm 0.05$ | $(82.85 \pm 0.28)$ | $81.04 \pm 0.09$ | $(75.71 \pm 0.33)$ | $72.87 \pm 0.06$ | $(72.96 \pm 0.14)$ |
| pFedMe | $85.58 \pm 0.23$ | $88.17 \pm 0.17$ | $79.33 \pm 0.14$ | $84.66 \pm 0.17$ | $72.11 \pm 0.23$ | $81.18 \pm 0.15$ |
| Ditto | $85.34 \pm 0.10$ | $87.55 \pm 0.09$ | $80.70 \pm 0.13$ | $83.39 \pm 0.12$ | $73.45 \pm 0.18$ | $80.08 \pm 0.20$ |
| FedRep | $(85.61 \pm 0.19)$ | $87.32 \pm 0.11$ | $(80.33 \pm 0.23)$ | $84.10 \pm 0.10$ | $(73.50 \pm 0.24)$ | $79.74 \pm 0.31$ |
| FedRoD | $86.02 \pm 0.12$ | $91.67 \pm 0.16$ | $\mathbf{81.31 \pm 0.15}$ | $85.91 \pm 0.15$ | $74.64 \pm 0.07$ | $81.37 \pm 0.17$ |
| FedBABU | $(85.67 \pm 0.24)$ | $91.34 \pm 0.19$ | $(79.57 \pm 0.23)$ | $83.22 \pm 0.33$ | $(73.88 \pm 0.19)$ | $80.58 \pm 0.22$ |
| Ours | $\mathbf{86.37 \pm 0.15}$ | $\mathbf{92.25 \pm 0.21}$ | $81.27 \pm 0.18$ | $\mathbf{86.59 \pm 0.17}$ | $\mathbf{75.52 \pm 0.11}$ | $\mathbf{82.69 \pm 0.16}$ |

Table 1: Average (3 trials) and standard deviation of the best test accuracies for generic/personalized models of various methods on CIFAR-10 with different non-IID settings. See also Remark 4.1.

| Methods | $\alpha = 1$ | | $\alpha = 0.1$ | |
|---------|------|------|------|------|
| | GM | PM | GM | PM |
| FedAvg | $48.37 \pm 0.22$ | $(52.64 \pm 0.48)$ | $38.61 \pm 0.27$ | $(39.27 \pm 0.42)$ |
| FedProx | $47.33 \pm 0.15$ | $(53.85 \pm 0.33)$ | $39.55 \pm 0.18$ | $(41.33 \pm 0.38)$ |
| FedDyn | $49.24 \pm 0.27$ | $(57.20 \pm 0.35)$ | $40.43 \pm 0.14$ | $(40.92 \pm 0.26)$ |
| FedAvgM | $48.55 \pm 0.19$ | $(55.60 \pm 0.26)$ | $39.03 \pm 0.08$ | $(40.85 \pm 0.19)$ |
| pFedMe | $47.29 \pm 0.27$ | $61.52 \pm 0.25$ | $38.22 \pm 0.23$ | $45.88 \pm 0.32$ |
| Ditto | $48.37 \pm 0.25$ | $60.47 \pm 0.27$ | $39.61 \pm 0.19$ | $43.12 \pm 0.28$ |
| FedRep | $(46.32 \pm 0.23)$ | $58.76 \pm 0.36$ | $(40.11 \pm 0.35)$ | $45.22 \pm 0.19$ |
| FedRoD | $50.07 \pm 0.16$ | $62.51 \pm 0.15$ | $40.58 \pm 0.22$ | $45.99 \pm 0.14$ |
| FedBABU | $(48.52 \pm 0.30)$ | $60.33 \pm 0.28$ | $(37.35 \pm 0.29)$ | $44.72 \pm 0.28$ |
| Ours | $\mathbf{51.39 \pm 0.22}$ | $\mathbf{63.15 \pm 0.16}$ | $\mathbf{40.67 \pm 0.14}$ | $\mathbf{46.82 \pm 0.23}$ |

Table 2: Average (3 trials) and standard deviation of the best test accuracies for generic/personalized models of various methods on CIFAR-100 with different non-IID settings. See also Remark 4.1.

### 3.5 Algorithm Summary

Our proposed spectral co-distillation framework combined with our wait-free local training protocol, is given in Algorithm 1. As an overview, we begin every communication round $t$ with the server broadcasting the global generic model $w_{\mathrm{G}}^{t-1}$ to each client for local computation. Each client $i$ would send back the updated generic model $w_{\mathrm{G},i}^{t}$ after $E_{\mathrm{G}}$ local computation steps for global model aggregation, then immediately start the personalized model training and continue until the global generic model $w_{\mathrm{G}}^{t}$ is received, which marks the start of the next communication round $t+1$.

**Remark on convergence analysis.** Note that the global loss function includes a weighted sum of the local loss functions and a regularizer. The regularizer is given in the form of the divergence function $\mathfrak{D}$, which is equivalent to KL divergence; cf. Sec. 3.2. As demonstrated in [50], the KL divergence usually exhibits convexity in terms of the model parameters. Consequently, since the model training undergoes (stochastic) gradient descent, it is possible to establish a convergence rate for the training of the global model (under the commonly employed assumption of smoothness of the local loss functions).

## 4 Experiments

### 4.1 Experiment setup

**Datasets, DNN models, federated settings, and evaluation metrics.** We evaluated our proposed PFL+ framework with $N$ clients on CIFAR-10/100 [51], and iNaturalist-2017, using model archi-

| Methods | FedProx | FedDyn | Ditto | FedRep | FedRoD | FedBABU | Ours |
|---------|---------|--------|-------|--------|--------|---------|------|
| GM | 39.46±0.39 | 39.35±0.27 | 39.33±0.33 | 39.81±0.41 | 40.16±0.35 | 39.23±0.53 | **41.75±0.37** |
| PM | 41.58±0.27 | 40.99±0.35 | 41.88±0.41 | 42.07±0.24 | 44.54±0.29 | 42.36±0.44 | **45.87±0.21** |

Table 3: Average (3 trials) and standard deviation of the best test accuracies for generic/personalized models of various methods on iNaturalist-2017 with non-IID setting $\alpha = 0.1$. See also Remark 4.1.

tectures ResNet-18/34 [52] and ResNet-50, respectively. For the experiments on CIFAR-10 (resp. CIFAR-100), we used $N = 100$ (resp. $N = 50$). For experiments on iNaturalist-2017 [53], we used $N = 20$. For dataset partition, we used the symmetric Dirichlet distribution to emulate real-world heterogeneous data distributions [9, 11], where the heterogeneity is controlled by the concentration parameter $\alpha$. (A smaller $\alpha$ indicates a higher degree of data heterogeneity.) For evaluation, we used two performance metrics:

- Generic model evaluation: global test accuracy (same metric in conventional FL).
- Personalized model evaluation: weighted average of local test accuracies.

For every client, the PM is evaluated on a local test set, whose underlying distribution is the same as that for the local training set. All the experiments are implemented with a full client participation scheme. Further experiment details, results on partial client participation, and the computation overhead discussion are provided in the Appendix.

**Remark 4.1.** For generic FL methods, personalized model (PM) accuracies are obtained by evaluating the generic model (GM) on local test sets. For PFL methods without GM training, GM accuracies are obtained by evaluating the averaged PM on the global test set.

**Baselines.** We compared our proposed method with the following SOTA PFL methods: pFedMe [17], Ditto [15], FedRoD [36], FedRep [24], and FedBABU [30]. Moreover, to have a fair performance evaluation of the generic models, we also include methods designed for conventional FL as baselines: FedAvg [19], FedProx [10], FedDyn [11], FedGen[54], and FedAvgM [9].

## 4.2 Performance comparison with state-of-the-art methods

We evaluated the generalizability of our proposed spectral co-distillation framework, as well as the communication cost performance of our wait-free training protocol for PFL+.

**Generalizability over heterogeneous settings.** We compared the best test accuracies with multiple baselines over the different levels of data heterogeneity, using the same system configuration. Tab. 1 and Tab. 2 give the main results on CIFAR-10 and CIFAR-100, respectively. In summary, our proposed framework achieves the best test accuracies across diverse heterogeneous data settings, outperforming all PFL and conventional FL baselines on both PM and GM test accuracies concurrently. We also investigated the performance on the real-world dataset iNaturalist2017 in Tab. 3, where our proposed method also achieves the best GM/PM test accuracies. We attribute such consistent outperformance to the bi-directional co-distillation design.

This demonstrates that: a) the spectral information of the generic model is useful for knowledge distillation during personalized model training; and b) using truncated spectral information of the personalized models could boost the performance of the generic model via careful spectrum truncation. (See Appendix for a sensitivity analysis of the truncation ratio $\tau$ and other hyper-parameters.)

**Communication cost comparison.** To demonstrate the superiority of the wait-free training protocol (**WF**), we evaluated the communica-

| Methods | 3 epochs | | 5 epochs | |
|---------|----------|----------|----------|----------|
| | 40% | 80% | 40% | 80% |
| | Speedup | | | |
| Ours (w/ WF) | 1.82 × | 1.56 × | 2.21× | 1.85× |
| Ditto w/ WF | 1.97× | 1.38 × | 2.87× | 1.93× |
| FedRoD w/ WF | 1.75× | 1.54 × | 2.42× | 2.19× |

Table 4: Communication cost comparison of various methods for personalized model accuracies on CIFAR-10 to reach target accuracy (40%/80%) with non-IID setting $\alpha = 0.1$. The speedup factors are with respect to the performance of the corresponding methods without WF.

tion cost performance of SOTA methods with/without the protocol on non-IID CIFAR-10 ($\alpha = 0.1$), in terms of the total runtime $\zeta_{\text{total}}$ for PM to reach the target test accuracy (40%/80%). A smaller $\zeta_{\text{total}}$ indicates higher communication efficiency. For PFL methods that train generic and personalized models using the compute-and-wait local training protocol, we evaluated Ditto and FedRoD. We conduct experiments with different numbers of epochs for local PM training (3 or 5 epochs). As shown in Tab. 4, our proposed wait-free training protocol could significantly improve the efficiency of convergence time and has the potential to boost the time efficiency of PFL+ methods.

| Method | $\alpha = 1$ | | $\alpha = 0.1$ | |
|---|---|---|---|---|
| | GM | PM | GM | PM |
| Ours | **86.37 ± 0.15** | **92.25 ± 0.21** | **75.52 ± 0.11** | **82.69 ± 0.16** |
| Ours w/o SCD-GM | 85.35 ± 0.11 | 91.86 ± 0.17 | 73.51 ± 0.17 | 81.03 ± 0.20 |
| Ours w/o SCD-PM | 82.74 ± 0.39 | 79.65 ± 0.83 | 68.96 ± 0.47 | 70.51 ± 1.21 |
| Ours w/o Both | 85.35 ± 0.11 | 79.65 ± 0.83 | 73.51 ± 0.17 | 70.51 ± 1.21 |

Table 5: Ablation study results on non-IID CIFAR-10 (average and standard deviation of 3 trials). **SCD-GM** (resp. **SCD-PM**) represents the spectral distillation approaches adopted during the training of generic (resp. personalized) model.

## 4.3 Ablation results

**Ablation study.** In our proposed spectral co-distillation framework, we introduce the bi-directional spectrum knowledge distillation to bridge the training of generic and personalized models with the target for training good generic and personalized models simultaneously. To achieve the target, truncated and full model spectrum information are adopted in different training stages. Here, we conduct an ablation study to evaluate the effectiveness of these two components (see Tab. 5 for the effects of each component), in which we apply the distillation approaches in the two training stages separately. In the setup where both SCD-PM and SCD-GM are removed (Case I), the GM training is identical to FedAvg. In the case of removing only SCD-PM while keeping SCD-GM (Case II), each PM would be trained locally without any knowledge distilled from the GM. This is akin to the client training its model by itself, separately from the server. Naturally, the PM performance would be drastically lower. As SCD-GM is kept in Case II, where the GM is the student and the PM is the teacher, since the PM's performance is drastically lower, we would expect a drop in the GM's performance. Informally, the model would be worse off with the distillation of bad knowledge, than without distillation.

As demonstrated in Tab. 5, both the distillation methods can boost the accuracy performance of generic and personalized models, whereas the bi-directional distillation can bridge the training performance of the generic and personalized models. Specifically, we can observe that, the SCD-PM module effectively transfers the knowledge from the generic model to the personalized model and avoids over-fitting during local training.

**Generalizability on new joining clients.** In a real-world PFL system, dynamic client participation should be regarded as an important factor to consider during algorithm design, in which there would be continually new clients joining the system during training. The PFL system needs to rapidly train a good personalized model that could generalize well on the new client's local data. To evaluate the generalizability of the system, we simulate a dynamic participation system with 80 in-training clients and 20 new clients on CIFAR-10 (partitioned

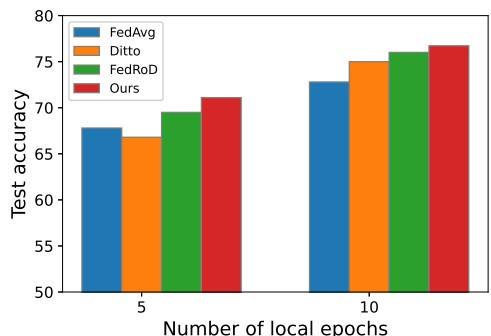

Figure 3: Performance comparison for generalizability on new clients of various methods.

by the Dirichlet distribution with $\alpha = 0.1$), and deal with new clients with the global model-based fine-tuning approach. Fig. 3 gives the results of the average test accuracies of the new clients. Among all evaluated methods, our method has the best average test accuracies, illustrating the fast adaptive capability of our method.

## 5  Conclusion

In this work, we propose a spectral co-distillation framework for PFL to learn better generic and personalized models simultaneously. As far as we know, this is the first work in PFL that represents the (dis-)similarity of models via their Fourier spectra. Even without co-distillation, there have been no other works that explore spectral distillation in PFL (or even in FL). The advantage of this new approach is clear from our experiments: We achieved outperformance in both generic and personalized model training. Our framework also incorporates a simple yet effective wait-free local training protocol to reduce the overall local training time.

**Limitations.** Our proposed spectral co-distillation framework, as currently formulated, does not deal with stragglers and adversarial attacks. Their influence on performance would require further investigation. Also, our protocol assumes a synchronized network connection, which may not be practical for scenarios with large system/network heterogeneity. Moreover, it would be good to consider a more realistic local training protocol design that takes into account the issues of network/system heterogeneity; we leave the extension as future work.

## Acknowledgements

This work is supported by the National Research Foundation, Singapore under its AI Singapore Program (AISG Award No: AISG-RP-2019-015), and by Ministry of Education, Singapore, under its Tier 2 Research Fund (MOE-T2EP20221-0016), and under its SUTD Kickstarter Initiative (SKI 2021_03_01), in part by the National Research Foundation, Singapore and Infocomm Media Development Authority under its Future Communications Research & Development Programme, in part by the National Natural Science Foundation of China under Grant 62271513, in part by the Zhejiang Lab Open Research Project (No. K2022PD0AB05), and in part by the Zhejiang–Singapore Innovation and AI Joint Research Lab.

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

# A    Appendix

In the appendix, we provide further details organized into the following parts:

- Sec. B introduces the implementation details for our proposed method with respect to the federated system settings, evaluation metrics, and the data partitions used in our experiments. In particular, we provide the implementation details for the experiments in "Generalizability on new joining clients".
- Sec. C gives further sensitivity analysis on the key parameters $\tau$, $\lambda_G$, and $\lambda_p$.
- Sec. D gives a discussion on the computational overhead.
- Sec. E gives a discussion on the differences and relations to asynchronized (personalized) federated learning (FL).
- Sec. F gives a discussion on how our wait-free local training protocol could be adapted for partial client participation.

# B    Implementation details

**Federated system settings.** All experiments were implemented using Pytorch. For all methods, we used an SGD local optimizer with a momentum of 0.5 and with no weight decay. We train all methods over a total number of $T = 500$ global communication rounds. Batch size for CIFAR-10/100 [51] and iNaturalist-2017 [53] are 10 and 128, respectively. As our aim is to train a personalized model for each client, we conducted the training using a full participation scheme, i.e., in every communication round, all clients would be selected to participate in training. For our proposed method, we used a learning rate of 0.01 (resp. 0.003) for both $\eta_G$ and $\eta_p$ when training on CIFAR-10/100 (resp. iNaturalist-2017).

**Details on the data partition.** We evaluated the performance of all methods with non-IID data partition. We simulated a non-IID data distribution via the symmetric Dirichlet distribution, in which we used the concentration parameter $\alpha$ to control data heterogeneity across the FL system [9, 11]. For the entire dataset $\mathcal{D}$ with $M$ classes, we sample a stochastic vector $\mathbf{d}_j$ with length $N$ (total number of clients) for each $1 \leq j \leq M$ via the Dirichlet distribution with common parameter $\alpha$. We shall treat $\mathbf{d}_j$ as the vector representing the portion of the data samples in class $j$ to be allocated to the clients. For example, if the second entry of $\mathbf{d}_1$ equals 0.05, then client 2 would be randomly allocated with 5% of the data samples from $\mathcal{D}$ in class 1. This allocation has no overlaps; two different clients have no common data samples. In particular, a smaller $\alpha$ indicates a higher degree of data heterogeneity, whereas a larger $\alpha$ will tend to generate identical-like distributions. (As $\alpha \to +\infty$, the data allocations become uniform.)

**Performance metrics.** The detailed calculation methods for the two metrics used in the evaluation are given as follows:

- We used the global test accuracy (same metric used in conventional FL) to evaluate the collaboratively trained global generic model, where the accuracy refers to the global test set (e.g., for CIFAR-10, the test set has 10,000 images).
- We used the weighted average of local test accuracies for personalized model evaluations, where each local test accuracy is obtained with respect to a local test set, and where the weights of the weighted average are the ratios of local dataset sizes $\frac{n_1}{n}, \frac{n_2}{n}, \ldots, \frac{n_N}{n}$. Let $acc_i$ denote the test accuracy for the personalized model at client $i$. Then the weighted average of local test accuracies is computed as $\sum_{i=1}^{N} \frac{n_i}{n} acc_i$. Note that for each client, the distribution of the local test set for each client is the same as the distribution of the local training set. Specifically, we perform sampling for the local test sets according to the distribution of training data for each respective client, where sampling is performed without replacement on the test set of the corresponding benchmark dataset, which is obtained via the Dirichlet distribution.

**Details for the experiments on new joining clients.** To evaluate the generalizability of our proposed federated co-distillation framework when there are new joining clients, we simulated a PFL system with 80 existing clients and 20 new clients on the CIFAR-10 dataset with non-IID data partition. In particular, we first partitioned the dataset into 100 local training sets via the Dirichlet distribution. We then performed training for the pre-determined 80 clients over 300 communication rounds for all methods, after which we conducted local fine-tuning [55, 23] on the new clients independently with a

certain number of local epochs based on the well-trained global generic model. (We tried 5 and 10 local epochs in our experiments.) The reported performance is the weighted average of the local test accuracies over the 20 new clients, in which the weights are the ratios of the dataset sizes relative to these 20 clients.

## C  Sensitivity analysis

Recall from Sec. 3.3 of the main paper that $\tau$ is the truncation ratio used to compute the truncated spectrum vectors. In our experiment results (as reported in the main paper), we used $\tau = 0.4$. To demonstrate the effect of this key hyperparameter, we conduct sensitivity analysis with multiple values of $0 < \tau \leq 1$ on two non-IID CIFAR-10 settings; see Tab. 6. $\tau = 0$ represents there is no spectral distillation for generic model training (which is equivalent to the setup "Ours w/o SCD-GM" in Tab. 5 of the ablation studies in the main paper ), while $\tau = 1$ indicates there is no spectrum truncation. In summary, $\tau = 0.4$ achieves the best performance for both generic model (GM) and personalized models (PM), which shows the effectiveness of the truncated spectrum in generic model training and implicitly demonstrates the characteristics of the low frequency of model Fourier spectrum.

| Method | $\alpha = 1$ | | $\alpha = 0.1$ | |
|---|---|---|---|---|
| | GM | PM | GM | PM |
| $\tau = 0$ | $85.35 \pm 0.11$ | $91.86 \pm 0.17$ | $73.51 \pm 0.17$ | $81.03 \pm 0.20$ |
| $\tau = 0.1$ | $85.18 \pm 0.26$ | $90.85 \pm 0.39$ | $72.66 \pm 0.11$ | $80.51 \pm 0.25$ |
| $\tau = 0.3$ | $85.42 \pm 0.22$ | $90.93 \pm 0.38$ | $72.59 \pm 0.38$ | $82.60 \pm 0.19$ |
| $\tau = 0.4$ | $\mathbf{86.37 \pm 0.15}$ | $\mathbf{92.25 \pm 0.21}$ | $\mathbf{75.52 \pm 0.11}$ | $\mathbf{82.69 \pm 0.16}$ |
| $\tau = 0.5$ | $86.33 \pm 0.08$ | $92.04 \pm 0.29$ | $74.91 \pm 0.31$ | $82.50 \pm 0.36$ |
| $\tau = 0.7$ | $86.08 \pm 0.43$ | $91.79 \pm 0.44$ | $72.85 \pm 0.28$ | $81.44 \pm 0.43$ |
| $\tau = 0.9$ | $86.25 \pm 0.39$ | $92.07 \pm 0.28$ | $72.57 \pm 0.31$ | $82.09 \pm 0.37$ |
| $\tau = 1.0$ | $85.47 \pm 0.26$ | $90.33 \pm 0.41$ | $73.55 \pm 0.17$ | $81.36 \pm 0.22$ |

Table 6: Sensitivity analysis for hyper-parameter $\tau$ (i.e., the truncation ratio for spectrum truncation, used in generic model training) on non-IID CIFAR-10 (average and standard deviation of 3 trials).

We also provide experiment results for the sensitivity analysis of the coefficients $\lambda_G$ and $\lambda_p$ on CIFAR-10 with data setting $\alpha = 1$ in Tab. 7 and Tab. 8.

| Setup | 0.01 | 0.05 | 0.1 | 0.3 |
|---|---|---|---|---|
| GM | 84.76 | 86.37 | 86.20 | 83.55 |
| PM | 90.33 | 92.25 | 91.35 | 89.64 |

Table 7: Sensitivity analysis for hyper-parameter $\lambda_G$ on non-IID CIFAR-10.

| Setup | 0.01 | 0.05 | 0.1 | 0.3 |
|---|---|---|---|---|
| GM | 86.37 | 86.33 | 85.61 | 85.72 |
| PM | 92.25 | 91.89 | 89.47 | 89.33 |

Table 8: Sensitivity analysis for hyper-parameter $\lambda_p$ on non-IID CIFAR-10.

## D  Computational overhead of the proposed method

In our proposed method, the additional computational overhead is dominated by the computation of spectral vectors, when compared to other model similarity/difference-based PFL methods [15, 17]. In our experiments, we use the famous fast Fourier transform (FFT) algorithm to compute the discrete Fourier transform of an input (complex) vector. Informally, the FFT algorithm is an efficient and

widely used algorithm that converts an input signal from the original domain into a representation in the frequency domain.

In our framework, when given an input weight vector $w_{p,i}^t \in \mathbb{R}^d$ (i.e. there are a total of $d$ model weights), its associated spectral vector $s(w_{p,i}^t)$ is computed using the FFT algorithm, with corresponding computational complexity $\Theta(d \log d)$; see, e.g., [56].

# E    Discussion on the differences/relations to asynchronized FL

Our main goal for proposing the wait-free PFL+ training protocol is to reduce the total running time overhead and thereby increase the training time efficiency. It should be noted that our wait-free local training protocol is synchronous, meaning that all the clients perform local computing, pushing generic model gradients, as well as generic model updates, in a synchronized manner. Similar to [48], our proposed protocol allows local computing during global communication.

In a similar context of reducing the total running time overhead, some asynchronized FL (AFL) methods have been proposed. We shall clarify that our wait-free local training protocol is vastly different from existing AFL methods, from the aspects of both the motivation and the communication-computation design. Specifically, AFL seeks to tackle the challenge of slow client in a heterogeneous FL system by reducing idle time, while in contrast, our method adopts a joint communication-computation design, thereby eliminating idle time. In particular, the usual idle waiting time has been replaced by training time for the personalized model in our proposed method. Since our method fully utilizes the idle waiting time at the local side with personalized model training, we did not consider incorporating the conventional AFL set-up into our framework.

To better demonstrate the outperformance of our wait-free (WF) local training protocol, we evaluated the performance of two state-of-the-art AFL methods (FedAsync [57] and FedBuff [58]) combined with the compute-and-wait PFL+ system using Ditto [15] (the PFL system that trains generic and personalized models simultaneously); see Tab. 9. The speedup factors are computed with respect to the performance of "Vanilla Ditto", where a larger speedup factor signifies better efficiency. The results demonstrate the outperformance of our proposed WF protocol in reducing the total running time overhead.

| Methods | 3 epochs | | 5 epochs | |
|---|---|---|---|---|
| | 40% | 80% | 40% | 80% |
| | Speedup | | | |
| Vanilla Ditto | $1\times$ | $1\times$ | $1\times$ | $1\times$ |
| Ditto w/ Our WF | $\mathbf{1.97\times}$ | $\mathbf{1.38\times}$ | $\mathbf{2.87\times}$ | $\mathbf{1.93\times}$ |
| Ditto w/ FedAsync | $1.36\times$ | $1.15\times$ | $1.29\times$ | $1.22\times$ |
| Ditto w/ FedBuff | $1.40\times$ | $1.26\times$ | $1.23\times$ | $1.40\times$ |

Table 9: Communication cost comparison of various methods for personalized model accuracies on CIFAR-10 to reach target accuracy (40%/80%) with non-IID setting $\alpha = 0.1$. The speedup factors are with respect to the performance of the Vanilla Ditto with the compute-and-wait protocol.

# F    Discussion on the applicability to the partial client participation scheme

In this work, we consider a full client participation scheme. However, in general, we may want the option that an edge device/client could be not selected to participate in training for numerous consecutive communication rounds, especially in the case of an FL system with a large number of clients. In this subsection, we discuss how our wait-free local training protocol should be extended/modified.

For an unselected client, the client should not be continuously training its personalized model using the "stale" GM model weights, which could result in model overfitting on limited data, and model drift/divergence.

To address this issue, there are two straightforward solutions:

- The first solution is to adopt a broadcasting scheme with regular latest GM broadcasts by the server. Such a scheme would ensure that every client receives the latest GM weights, thereby alleviating the negative impact of stale GM weights for unselected clients, and avoiding potential client drift. In real-world network connections, the bandwidth of the downlink connection from the server to the clients is typically larger than the corresponding bandwidth of the uplink connection. Hence, such design would be practical.
- The second solution is to set an upper limit on the number of local training steps. The basic underlying idea is to terminate the local PM training after a certain number of local training steps. This could be an alternative option to address the issue of limited communication for cross-device FL scenarios. It should be noted that such design would still reduce the overall total training time, since the PM training is still performed within the idle waiting time in conventional FL.

We give further experiment results (see below) for partial participation with two typical participation rates ($c$=0.1 and 0.2) on non-IID CIFAR-10 with $\alpha = 1$ in Tab. 10.

| Methods | $c$=0.1 | | $c$=0.2 | |
| --- | --- | --- | --- | --- |
| | GM | PM | GM | PM |
| FedAvg | 83.16 | 78.25 | 83.84 | 79.03 |
| FedProx | 82.97 | 78.23 | 83.75 | 79.44 |
| FedRoD | 84.22 | 89.31 | 84.90 | 89.68 |
| Ours | 84.35 | 89.56 | 85.11 | 90.37 |

Table 10: Sensitivity analysis for hyper-parameter $\lambda_G$ on non-IID CIFAR-10.

These results show that our method is applicable for partial participation, and even outperforms our baselines.

