# OpenReview forum: "Spectral Co-Distillation for Personalized Federated Learning"
_NeurIPS.cc/2023/Conference — NeurIPS 2023 poster_

### Official Review · Reviewer_jcpB · 2023-07-04

**Soundness:** 3 good
**Presentation:** 4 excellent
**Contribution:** 4 excellent
**Rating:** 7
**Confidence:** 3

**Summary:**

The authors proposed a personalized federated learning (PFL) method, where the (dis-)similarity of models are represented through their Fourier spectra. The authors also proposed a bi-directional knowledge distillation between the generic and personalized models and a wait-free local training protocol that can utilize the idle time during global communication so as to reduce the total runtime. Extensive experiments and ablation studies are conducted to demonstrate the performance.

**Strengths:**

1. The authors are the first to characterize (dis-)similarity of models via Fourier spectra in PFL, and the proposed bi-directional knowledge distillation is novel. Further, the proposed simple wait-free local training protocol seems to have the potential of speeding up many existing PFL methods.
2. The paper is well organized and clearly written.
3. The experiment results seem strong.

**Weaknesses:**

The use of Fourier spectra is motivated by previous studies on deep learning theories. As it stands, the method seems like a mathematically motivated heuristic. It would be excellent if certain convergence guarantees or error bounds can be proved, even if just for the most vanilla case.

**Questions:**

1. What are some limitations of this wait-free local training protocol? What kind of methods can adaptively include this protocol into their framework?
2. In the case that a FL system has a large number of edge devices/clients, the participation rate may be low. If I’m understanding correctly, the personalized models keep on training until the updated global weights are received. If an edge device doesn’t participate in training for a long time, wouldn’t this protocol incur a lot of local computation? Also, if the personalized models keep on training without the most updated version of the global weights, is it possible that the personalized models will drift away from the rest of the system?

**Limitations:**

The authors have adequately addressed the limitations. Their work does not seem to have any potential negative societal impact.

---

> ### Author Rebuttal · Authors · 2023-08-10
>
> We are very encouraged by the reviewer's positive feedback. Below, we address the comments point-by-point.
>
> > The use of Fourier spectra is motivated by previous studies on deep learning theories. As it stands, the method seems like a mathematically motivated heuristic. It would be excellent if certain convergence guarantees or error bounds can be proved, even if just for the most vanilla case.
>
> We thank the reviewer for the thoughtful comment.
> We agree that the theoretical analysis would benefit the significance of the proposed method. However, given the word limit and limited time for rebuttal, we could not include a complete convergence analysis in this response.
>
> That said, we can stretch the approaches for an analysis of the convergence rate under the proposed framework, as follows: First, note that the global loss function includes a weighted sum of the local loss functions and a regularizer, whereas the latter is given in the form of a KL divergence. As demonstrated in [1], the KL divergence usually exhibits convexity in terms of the model parameters. Consequently, noting that the model training undergoes (stochastic) gradient descent, we can establish a convergence rate for the global part (under the commonly employed assumption of smoothness of the local loss functions). The convergence rate of the local training follows similar approaches.
>
> In particular, the intricate operations involved in our proposed framework will be reflected in the multiplicative factors in the convergence rate. We shall highlight this and provide the analysis in our revised supplementary material.
>
> [1] How Does Loss Function Affect Generalization Performance of Deep Learning? Application to Human Age Estimation
>
>
>
> > What are some limitations of this wait-free local training protocol? What kind of methods can adaptively include this protocol into their framework?
>
> 1) Limitations of our wait-free training protocol are described in lines 311-312 of the main paper, and also in lines 78-80 of the supplementary material.
>    Essentially, our protocol assumes a synchronized network connection, which may not be practical for scenarios with large system/network heterogeneity.
>    We shall leave the extension to the case of asynchronous PFL as our future work.
>
> 2) This protocol is designed for PFL+ (see line 130 of our main paper), in which a generic model (GM) and $N$ oersonalized models (PMs) are trained. Our protocol could easily be incorporated into any methods for PFL+ where the training of the GM and PMs can be decoupled into a sequential process, such as Ditto [1] and FedRoD [2]. In particular, in Table 2 of the supplementary material,
>    we have shown how our wait-free local training protocol, when applied to Ditto, gives a significant reduction in the total running time overhead.
>
> [1] Ditto: Fair and Robust Federated Learning Through Personalization
>
> [2] On Bridging Generic and Personalized Federated Learning for Image Classification
>
> > In the case that a FL system has a large number of edge devices/clients, the participation rate may be low. If I’m understanding correctly, the personalized models keep on training until the updated global weights are received. If an edge device doesn’t participate in training for a long time, wouldn’t this protocol incur a lot of local computation? Also, if the personalized models keep on training without the most updated version of the global weights, is it possible that the personalized models will drift away from the rest of the system?
>
> We thank the reviewer for the thoughtful comments, which cover the key considerations for how our wait-free local training protocol should be extended/modified.
>
> For partial client participation over an FL system with a large number of clients, it is possible that an edge device/client would not be selected to participate in training for numerous consecutive communication rounds.
> For such an unselected client, the client should not be continuously training its personalized model using the ``stale'' GM model weights, which as the reviewer has aptly pointed out, could result in model overfitting on limited data, and model drift/divergence.
>
> To address this issue, there are two straightforward solutions:
>
> i) The first solution is to adopt a broadcasting scheme with regular latest GM broadcasts by the server. Such a scheme would ensure that every client receives the latest GM weights, thereby alleviating the negative impact of stale GM weights for unselected clients, and avoiding potential client drift.
> In real-world network connections, the bandwidth of the downlink connection from the server to the clients is typically larger than the corresponding bandwidth of the uplink connection. Hence, such design would be practical.
>
> ii) The second solution is to set an upper limit on the number of local training steps. The basic underlying idea is to terminate the local PM training after a certain number of local training steps. This could be an alternative option to address the issue of limited communication for cross-device FL scenarios. It should be noted that such design would still reduce the overall total training time, since the PM training is still performed within the idle waiting time in conventional FL.
>
> We will include the above elaboration in our revision, both in our main paper and in the supplementary material.

---

> > ### Comment · Reviewer_jcpB · 2023-08-17
> >
> > I appreciate the authors’ thorough answer, and I believe the proposed revisions would make the paper stronger.

---

> > > ### Author Response · Authors · 2023-08-21
> > >
> > > We thank the reviewer very much for appreciating our proposed revisions. Also, we thank the reviewer once again for the thoughtful comments that led to our proposed revisions. We will include the above discussions in our revised paper.

---

### Official Review · Reviewer_6pBd · 2023-07-08

**Soundness:** 3 good
**Presentation:** 3 good
**Contribution:** 3 good
**Rating:** 6
**Confidence:** 3

**Summary:**

This paper proposes a novel approach to address the challenge of data heterogeneity in personalized federated learning (PFL) by introducing a spectral co-distillation framework based on model spectrum information. The proposed method combines a wait-free local training protocol with bi-directional spectrum knowledge distillation to bridge the training of generic and personalized models. The authors evaluate their approach on CIFAR-10/100 and iNaturalist-2017 datasets using ResNet-18/34 and ResNet-50 architectures, and show that it outperforms existing PFL and conventional federated learning baselines on both personalized and generic model test accuracies. The proposed wait-free training protocol also significantly improves the efficiency of convergence time and has the potential to boost the time efficiency of PFL+ methods. The authors conclude that their approach demonstrates the usefulness of spectral information for knowledge distillation during personalized model training and the potential of using truncated spectral information of personalized models to boost the performance of the generic model.

**Strengths:**

The proposed spectral co-distillation framework is a novel approach that addresses the challenge of data heterogeneity in personalized federated learning.
The wait-free local training protocol significantly improves the efficiency of convergence time and has the potential to boost the time efficiency of PFL+ methods.
The proposed approach outperforms existing PFL and conventional federated learning baselines on both personalized and generic model test accuracies across multiple datasets and architectures.
The authors provide a sensitivity analysis of the truncation ratio τ and demonstrate the usefulness of spectral information for knowledge distillation during personalized model training.


**Weaknesses:**

The proposed approach may require additional computational resources to compute the spectral information of the models, which could increase the overall training time and cost.
The authors do not provide a comprehensive analysis of the privacy and security implications of the proposed approach, which could be a concern in real-world applications.
The authors only evaluate their approach on a limited set of datasets and architectures, and it is unclear how well it would generalize to other domains and settings.

**Questions:**

see above

**Limitations:**

see above

---

> ### Author Rebuttal · Authors · 2023-08-10
>
> We thank the reviewer for the overall positive feedback. Below we address the highlighted weakness point-by-point.
>
> > The proposed approach may require additional computational resources to compute the spectral information of the models, which could increase the overall training time and cost. The authors do not provide a comprehensive analysis of the privacy and security implications of the proposed approach, which could be a concern in real-world applications. The authors only evaluate their approach on a limited set of datasets and architectures, and it is unclear how well it would generalize to other domains and settings.
>
>
>
> **1. Computational resource requirement**
> We agree that our proposed spectral distillation process would require additional computational resources;
> this was discussed in Section 3 of the supplementary material. Nonetheless, our proposed wait-free training protocol could  mitigate the impact of additional running time via an effective utilization of the idle time for PM training.
>
>
>
> **2. Privacy and security implications**
> Spectrum vectors and their truncated versions are computed solely on the model weights.
> In FL, since model weights are already transmitted between the server and the clients,
> the computation of such spectrum vectors maintains the same level of privacy and security already presented in FL.
>
>
>
> **3. Experiments**
> To recap, we have provided experimental results on three datasets, and on different architectures with diverse non-IID data settings, including an evaluation on iNaturalist-2017, a large real-world image dataset. Such experiment setups are typically chosen in existing FL works, so we adopt the same setup for a fairer comparison with the baselines. As suggested, we have conducted additional experiments on a new task over an FL-oriented benchmark data setup, FEMNIST in pFL-Bench [1]. The results are given as follows.
>
> We compare both the accuracy and its standard deviation across clients during averaging as a fariness-related metric for PM performance. We used 200 clients and the same data setup as given in the pFL-Bench paper.
>
> | Methods | $Acc$ | $\sigma$ |
> | ------- | ----- | -------- |
> | FedAvg  | 82.35 | 6.23     |
> | FedProx | 81.87 | 5.92     |
> | Ditto   | 85.26 | 5.04     |
> | FedRoD  | 86.03 | 7.36     |
> | Ours    | 87.51 | 6.77     |
>
>
> We hope our outperformance in this additional real-world setup would address the reviewer's concerns.
> A comprehensive comparison will be given in our revised supplementary material.
>
> [1] pFL-Bench: A Comprehensive Benchmark for Personalized Federated Learning

---

### Official Review · Reviewer_UBhD · 2023-07-09

**Soundness:** 3 good
**Presentation:** 2 fair
**Contribution:** 2 fair
**Rating:** 5
**Confidence:** 3

**Summary:**

This paper introduces a novel spectral co-distillation framework for PFL, leveraging a Fourier spectrum perspective to measure model similarities. With the assumption that local models share low-frequency features and the global model's Fourier spectrum includes averaged high-level features from all clients, the authors propose a bi-directional knowledge distillation approach to improve performance on both sides. Additionally, a wait-free local training protocol is introduced to reduce runtime. Experimental results on multiple datasets with diverse heterogeneous data settings validate the effectiveness of the framework.

**Strengths:**

**Novel Aspect of Measuring Model Similarity**: The novel frequency perspective for measuring model similarity is intriguing and holds promise for producing a more principled solution for PFL. This aspect deserves encouragement and further exploration.

**Interesting Motivation**: The consideration of shared low-frequency components among personalized models and the generic model provides a unique perspective on describing their relationship.

**Weaknesses:**

**Lack of Theoretical Analysis**: The authors highlight in lines 39-41 that existing works lack theoretical support and heavily rely on empirical observations. However, the core motivation behind spectral distillation (line 177) is also intuitive, and no theoretical analysis or guarantees are provided in the paper. This limits the contribution of the proposed method.

**Insufficient Experiments**:
- The paper would benefit from empirical analysis of model spectra to justify or validate the core motivation. Specifically, it is important to investigate if the robustness of low-frequency components against uniform random perturbation, as demonstrated in Ref[23], remains applicable in the Non-IIDness setting of FL. This would support the assumption of low-frequency component sharing among local models. Providing experimental evidence would solidify and amplify the significance of this assumption.
- Empirical analysis for key factors $\lambda_p$ and $\lambda_G$ is missing.

**Questions:**

- The calculation process of the model spectrum is unclear based on the existing descriptions (line 151-155 in the main file, line 73-74 in the appendix). Clarifications are needed, similar to prior work [22][23]. It is better to provide detailed explanations of how the overall model parameters are organized into a single vector and how the spectral vector is calculated. Offering a specific example, such as AlexNet, would greatly enhance the understanding of these processes.
- The explanation for the results of the ablation study in Table 5 is inaccurate. Upon comparing the "w/o SCD-PM" line with the "w/o Both" line, it can be observed that solely adopting SCD-GM is detrimental to performance (GM's performance decreases for both $\alpha=1$ and $\alpha=0.1$, while PM's performance remains steady). However, when comparing the "Ours" line with the "w/o SCD-GM" line, the performance improves upon the further adoption of SCD-GM in addition to SCD-PM. This indicates that SCD-PM holds a higher priority and plays a more fundamental role.

**Limitations:**

A section specifically addressing the limitations and broad impact is missing.

---

> ### Author Rebuttal · Authors · 2023-08-10
>
> We thank the reviewer for the valuable feedback. Below we address the reviewer's concerns point-by-point.
>
> > [Lack of Theoretical Analysis] The authors highlight in lines 39-41 that existing works lack theoretical support and heavily rely on empirical observations. However, the core motivation behind spectral distillation (line 177) is also intuitive, and no theoretical analysis or guarantees are provided in the paper. This limits the contribution of the proposed method
>
> We thank the reviewer for the comment. For the lack of theoretical analysis of our framework, please see our response to Reviewer jcpB, which addresses what we shall include in our revised paper.
>
> > [Insufficient Experiments] The paper would benefit from empirical analysis of model spectra to justify or validate the core motivation. Specifically, it is important to investigate if the robustness of low-frequency components against uniform random perturbation, as demonstrated in Ref[23], remains applicable in the Non-IIDness setting of FL. This would support the assumption of low-frequency component sharing among local models. Providing experimental evidence would solidify and amplify the significance of this assumption.
>
> We thank the reviewer for this insightful suggestion.
> First, we agree that exploring the robustness of low-frequency components to random perturbations would amplify the significance of our proposed spectral co-distillation framework.
>
> As suggested, we have conducted additional experiments with the label noise provided in [1] on CIFAR-10 with data setting $\alpha=1$, where the label noise ratio is denoted by $\beta$. The experiment setup is the same as described in Sec. 3.1 of [1].
>
>
>
> | $\beta$ | 0     | 0.1   | 0.2   | 0.3   |
> | ------- | ----- | ----- | ----- | ----- |
> | GM      | 86.37 | 85.43 | 82.78 | 80.51 |
> | PM      | 92.25 | 89.36 | 85.38 | 82.11 |
>
> We should also note that a detailed analysis of the robustness of spectral-based methods for FL/PFL is not the main focus of this paper, but we would certainly be keen to explore this as future work.
>
> [1] Spectral Bias in Practice: the Role of Function Frequency in Generalization
>
> > Empirical analysis for key factors $\lambda_G$ and $\lambda_p$ is missing.
>
> We thank the reviewer for pointing this out. We have provided additional experiment results for the sensitivity analysis of the coefficients $\lambda_\text{G}$ and $\lambda_\text{p}$ on CIFAR-10 with data setting $\alpha=1$. A comprehensive sensitivity analysis will be provided in  our revised supplementary material.
>
> Sensitivity analysis of $\lambda_\text{G}$
>
> | $\lambda_\text{G}$ | 0.01  | 0.05  | 0.1   | 0.3   |
> | ------------------ | ----- | ----- | ----- | ----- |
> | GM                 | 84.76 | 86.37 | 86.20 | 83.55 |
> | PM                 | 90.33 | 92.25 | 91.35 | 89.64 |
>
> Sensitivity analysis of $\lambda_\text{p}$
>
> | $\lambda_\text{p}$ | 0.01  | 0.05  | 0.1   | 0.3   |
> | ------------------ | ----- | ----- | ----- | ----- |
> | GM                 | 86.37 | 86.33 | 85.61 | 85.72 |
> | PM                 | 92.25 | 91.89 | 89.47 | 89.33 |
>
>
>
> > The calculation process of the model spectrum is unclear based on the existing descriptions (line 151-155 in the main file, line 73-74 in the appendix). Clarifications are needed, similar to prior work [22][23]. It is better to provide detailed explanations of how the overall model parameters are organized into a single vector and how the spectral vector is calculated. Offering a specific example, such as AlexNet, would greatly enhance the understanding of these processes.
>
> We thank the reviewer for this suggestion. The calculation of the spectral vector directly follows the code provided by the spectral bias paper [1]. We will provide more details in our revised paper.
>
> [1] On the Spectral Bias of Neural Networks
>
> > The explanation for the results of the ablation study in Table 5 is inaccurate. Upon comparing the "w/o SCD-PM" line with the "w/o Both" line, it can be observed that solely adopting SCD-GM is detrimental to performance (GM's performance decreases for both $\alpha=1$ and $\alpha=0.1$, while PM's performance remains steady). However, when comparing the "Ours" line with the "w/o SCD-GM" line, the performance improves upon the further adoption of SCD-GM in addition to SCD-PM. This indicates that SCD-PM holds a higher priority and plays a more fundamental role
>
> Please note that our summarized explanation for the results of the ablation study in Table 5 is correct.
> Reviewer 8564 raised a similar question, and a detailed explanation can be found in our response to the last question from Reviewer 8564.

---

### Official Review · Reviewer_rUNY · 2023-07-18

**Soundness:** 3 good
**Presentation:** 1 poor
**Contribution:** 3 good
**Rating:** 5
**Confidence:** 4

**Summary:**

This paper proposes a method for PFL, which aims to address the challenge of data heterogeneity. The main idea is to use spectral distillation to capture the relations between the generic global and personalized local models based on the similarity of spectral information. The paper introduces a co-distillation framework that establishes a bi-directional bridge between generic and personalized model training and a wait-free local training protocol that utilizes the local idle time in conventional PFL. The paper evaluates the proposed method on two datasets with heterogeneous settings and shows that it outperforms other existing PFL methods.

**Strengths:**

The paper presents an interesting idea of using spectral distillation for PFL to share low-level features, which differs from existing methods relying on model weight similarity. The paper provides some intuition on why spectral distillation can better capture the generic versus personalized model representations. Furthermore, the paper proposes a wait-free local training protocol that can improve the communication efficiency of PFL, which is a general technique that can be adopted by other PFL methods.

**Weaknesses:**

1. The clarity of this paper should be improved. Please be careful with using uncommon abbreviations, or provide explanations where appropriate. GM and PM should be clarified in the Figure 1 caption; in Line 232, SM should be spelled out as Supplementary Material; In Table 4, what do 3/5 epochs mean?  I guess it is the local training steps.

2. Experiment settings need to be improved. The evaluation of PM and GM is not comparable, as they are completely different sets. It would be better to provide additional results of PM and GM on the same test set, such as the global test set. Moreover, the datasets used in the experiments are obtained by splitting with Dirichlet distribution, which are not real FL datasets. Please consider following the experimental setup in pFL-Bench [1], using datasets like FEMNIST [2], and reporting more fairness-related metrics, such as standard deviation.

3. The reproducibility of the experiments is questionable. I could not find how the local test set was generated in CIFAR-10, such as the train/test ratio, in the main text or the supplementary material.

[1] pFL-Bench: A Comprehensive Benchmark for Personalized Federated Learning, Advances in Neural Information Processing Systems 35 (NeurIPS 2022) Datasets and Benchmarks Track

[2] LEAF: A Benchmark for Federated Settings.

**Questions:**

Please refer to the Weaknesses section, and address the issues raised there. I will re-evaluate my assessment of this paper if these Weaknesses are resolved.

**Limitations:**

The author discusses the limitation of this paper.

---

> ### Author Rebuttal · Authors · 2023-08-10
>
> We thank the reviewer for the overall positive feedback. Below we address the highlighted weakness point-by-point.
>
> > The clarity of this paper should be improved. Please be careful with using uncommon abbreviations, or provide explanations where appropriate. GM and PM should be clarified in the Figure 1 caption; in Line 232, SM should be spelled out as Supplementary Material; In Table 4, what do 3/5 epochs mean? I guess it is the local training steps.
>
> We thank the reviewer for the suggestion to improve clarity. We will update the caption of Figure 1 with an explanation of the abbreviations (PM and GM). Our proposed new caption is given as follows:
>
> "Spectral co-distillation framework with wait-free local training for PFL, in which the generic model (GM) training and the personalized model (PM) training are carried out via spectral distillation in two different stages.''
>
> We have checked that there are only two instances of the abbreviation SM in our paper.
> Hence, we shall replace both instances with the full wording "supplementary material''.
>
> Regarding "3 or 5 epochs'' in Table 4, they both refer to the number of epochs for local PM training. We apologize for the missing explanation.
>
>
>
> > Experiment settings need to be improved. The evaluation of PM and GM is not comparable, as they are completely different sets. It would be better to provide additional results of PM and GM on the same test set, such as the global test set. Moreover, the datasets used in the experiments are obtained by splitting with Dirichlet distribution, which are not real FL datasets. Please consider following the experimental setup in pFL-Bench [1], using datasets like FEMNIST [2], and reporting more fairness-related metrics, such as standard deviation.
>
> We thank the reviewer for valuable comments on how our experiment setup could be improved. We first clarify that in the PFL+ scenario (see line 130 in main paper), which is the scope of our paper, the GM and PMs do not just have different training sets, but they also have different evaluation setups.
> For the results in Tables 1--3, we compare the results of different methods in a single column, rather than directly comparing them across GM and PMs.
> In particular, the evaluation of each PM is based on "personalized" data distribution. This means that testing all PMs on the same global test set would change the scope of our PFL+ work.
> Moreover, for the test set design, we adopted a setup that is identical to most PFL+ works [1] , in which each PM is evaluated on a local test set, whose underlying distribution is the same as that for the local training set.
>
> We thank the reviewer very much for pointing out the relevance of pFL-Bench [2] and FEMNIST in LEAF [3]. In our revision, we will cite both papers and provide further discussion on them. As requested, we have conducted a comparison on this pFL-Bench setup. (The full comprehensive comparison will be given in our revised paper and supplementary material.)
> For the performance of PMs, we evaluated both the test accuracy and standard deviation across clients during averaging, where this standard deviation serves as a fairness-related metric. For these experiments, we used 200 clients and the same data setup as given in the pFL-Bench paper.
>
> | Methods | $Acc$     | $\sigma$ |
> | ------- | --------- | -------- |
> | FedAvg  | 82.35     | 6.23     |
> | FedProx | 81.87     | 5.92     |
> | Ditto   | 85.26     | 5.04     |
> | FedRoD  | 86.03     | 7.36     |
> | Ours    | **87.51** | 6.77     |
>
> [1] FedMask: Joint Computation and Communication-Efficient Personalized Federated Learning via Heterogeneous Masking
>
> [2] pFL-Bench: A Comprehensive Benchmark for Personalized Federated Learning
>
> [3] LEAF: A Benchmark for Federated Settings
>
>
>
> > The reproducibility of the experiments is questionable. I could not find how the local test set was generated in CIFAR-10, such as the train/test ratio, in the main text or the supplementary material.
>
> Note that in our supplementary material, we have provided a description of the local test set sampling (see lines 40-44).
> However, we agree that the details on the generation of local test sets could be made clearer.
> Overall, the local test set for each client is sampled from the test set of the corresponding benchmark dataset, such as the 10000 test images in CIFAR-10. Thus, we plan to revise the last sentence (lines 42-44) in the supplementary material to the following:
>
> "Specifically, we perform sampling for the local test sets according to the distribution of training data for each respective client, where sampling is performed without replacement on the test set of the corresponding benchmark dataset, which is obtained via the Dirichlet distribution.''
>
> We hope our proposed revised wording would improve clarity.

---

> > ### Comment · Reviewer_rUNY · 2023-08-21
> >
> > I have reviewed the revised manuscript and the author's responses to the other reviewers' comments. I acknowledge that the authors have made some efforts to improve the presentation and clarity of their work. However, I can't find the full comprehensive comparison in FEMNIST in the revised manuscript as the author promised. Therefore, I will maintain my original score.

---

> > > ### Author Response · Authors · 2023-08-21
> > >
> > > We thank the reviewer very much for the feedback to our rebuttal. We would like to clarify that according to the requirements of NeurIPS 2023, we cannot post our revised manuscript during the rebuttal period. Please refer to the 3rd and 4th points of the Reviewing/Discussion process on the **NeurIPS 2023 FAQ For Authors** page.
> > >
> > > Regarding the additional experiments on FEMNIST based on pFL-Bench and LEAF, which we have included in our rebuttal (see previous post), we would like to highlight that due to the character limit and limited time for rebuttal, the FEMNIST experiment results in our rebuttal are not the final version.
> > > In our revision, we will provide a comprehensive analysis for FEMNIST on different client sampling rates
> > > as used in pFL-Bench.
> > > We will properly acknowledge and cite both pFL-Bench and LEAF.
> > >
> > > We hope the above clarifications could help the reviewer in re-evaluating the assessment.

---

### Official Review · Reviewer_FUrR · 2023-07-21

**Soundness:** 3 good
**Presentation:** 3 good
**Contribution:** 2 fair
**Rating:** 4
**Confidence:** 3

**Summary:**

This paper introduces a spectrum based co-distillation for personalized federated learning; which involves the local model training with regularization/distillation with respect to global parameters's spectral representation and the global model training with regularization/distillation with respect to local parameters's spectral representation.

**Strengths:**

1. Personalized FL is an important and timely topic
2. The approach using spectral transformation for distillation is novel and interesting
3. The experimental approaches show the effectiveness of the proposed approach

**Weaknesses:**

1. The technical contribution is relatively low
2. The experimental results used only two datasets, not very convincing.

**Questions:**

1. Can you highlight both the winner and the second in Table 1? When $\alpha=0.5$, the winner should be FedRod?

2. Why other methods should be affected by $\alpha$ which is a parameter in your model?

3. What are the results if you don't employ spectral based distillation?

**Limitations:**

Yes

---

> ### Author Rebuttal · Authors · 2023-08-09
>
> We thank the reviewer for the comments.
>
> > The technical contribution is relatively low
>
> We have the following novel technical contributions, which have also been pointed by all the other reviewers.
>
> 1. We are the first to characterize model (dis-)similarity of personalized models (PMs) versus the generic model (GM) in PFL via fourier spectra of models. This provides a new perspective for modelling the relationship between the PMs and GM.
>
> 2. We propose a co-distillation framework to bridge the training of the GM and PMs in a two-way manner.
>
> 3. We proposed a wait-free training protocol to reduce the total running time in PFL, which could also be easily incorporated into other PFL frameworks.
>
>
> > The experimental results used only two datasets, not very convincing.
>
> For our experiments, we evaluated the performance of our method over **three** different datasets (**not two**), CIFAR-10 in Table 1, CIFAR-100 in Table 2, and iNaturalist-2017 in Table 3, and with different heterogeneous data settings to demonstrate the efficacy and outperformance. Moreover, in this rebuttal, we have provided further experiment results on an additional 4th dataset FEMNIST. See response to Reviewer rUNY.
>
>
>
> > Can you highlight both the winner and the second in Table 1? When $\alpha=0.5$, the winner should be FedRod?
>
> We thank the reviewer for this comment. We will highlight the top 2 accuracies for Table 1 in our revision.
> We apologize for the typo for the highest GM accuracy under $\alpha=0.5$ in Table 1; this should be FedRoD.
>
>
>
> > Why other methods should be affected by $\alpha$ which is a parameter in your model?
>
>
> The parameter $\alpha$, which is used throughout all experiments, is the concentration parameter for the symmetric Dirichlet distribution, used to control data heterogeneity across the clients in the FL system. Full description and details can be found in lines 224-227 of the main paper, and also in lines 20-30 of the supplementary material. Note that different $\alpha$ values would correspond to different levels of data heterogeneity. There is extensive empirical evidence in the FL literature that a higher level of data heterogeneity would result in a higher degree of performance degradation. Therefore, all the methods would be affected by $\alpha$.
>
>
>
> > What are the results if you don't employ spectral based distillation?
>
> Please see **Table 5** of our paper. In this table, we provide an ablation study that includes the case of not using spectral distillation. Specifically, we show that, without spectral distillation at the GM and/or without spectral distillation at the PMs, there is a drop in performance.
> Hence, spectral distillation (both for GM and PMs) is effective and crucial for the outperformance of our method over the baselines.

---

### Official Review · Reviewer_8564 · 2023-07-27

**Soundness:** 2 fair
**Presentation:** 2 fair
**Contribution:** 2 fair
**Rating:** 5
**Confidence:** 4

**Summary:**

1. This paper proposes a new personal FL methodology: FedSCD. The authors facilitates the generic and personal models to capture global/local distributions through spectral distillation. A co-distillation framework is introduced to help generic and personal models to learn from each other. From the Fourier analysis, since the DNN tends to learn the low frequency features before the high frequency features, the feature level in model representation can be obtained from Fourier spectrum. According to the author, the global model can learn from some low-level features that are commonly shared among diverse local models, and the averaged high level features can benefit the personalized model.
2. Instead of regularizing the local model not to diverge too much from the global model, the spectrum distillation minimizes the loss from the KLDivergence between the spectrum vector of generic and personal model.
3. The paper also designs a wait-free local training protocol that aims to improve the training efficacy by paralleling the process of server aggregation and local personalized model training.

**Strengths:**

1. The idea that utilizing Fourier spectrum information to seek for common global model architecture is novel and properly aligns with the personalization Federated Learning scheme
2. The distillation process, compared to other knowledge distillation methods, does not require public data access on server side, which is more applicable to real-world setting.

**Weaknesses:**

[Methodology Setup and Justification]
1. Co-spectrum distillation design.

The design of personal model spectrum distillation is somehow counter-intuitive. In line 56-58, the author says “Dually, the entire Fourier spectrum of the generic model, which includes the "averaged" high-level features across all clients, would benefit the training of the personalized models.” If the low frequency feature is shared among all clients and should be learned by the generic model, the personalized models should have different high frequency features since each clients have distinct data distribution/scenario that they need to adapt on. Then how can the averaged high-leave features benefit client personalization?

However, empirically, the spectrum distillation on personal model greatly improves the performance, should such improvement be credited from the fact that during the spectrum distillation, the commonly shared low frequency features in personal model generalized better? So the improvement is actually from better trained low frequency features instead of averaged high-level features? I would like to see author provides more theoretical adjustments on this.

2. Value of Wait-free protocol.

The wait-free protocol can be easily implemented under personal FL framework and effectively reduce the communication cost, but it is off-track from the main topic of the paper: co-spectrum distillation. Also, the effectiveness of the proposed wait-free protocol is mainly based on the assumption of all clients share similar training speed. If the fastest client and slowest client has large gap in training speed, the fastest client still need wait for the slowest client. The problem of heterogeneous hardware/computation power is also under explore, and many works have made contribution on this topic[1]. If reducing the communication cost is also a main focus of the paper, the wait-free protocol needs better design to become a new methodology.

3. Missing Fourier Spectrum Analysis in Neural Networks in Related Work.

Spectrum analysis of neural networks is apparently a key component in the proposed methodology, but not everyone is familiar with this topic. It will be nice to include a short description of Fourier spectrum in neural networks.

---

[Experiment Setting]
Overall, the experiment needs better simulation to mimic real settings. The flaws of experiment are below:
1. client participation rate.

For all experiment, all clients participate in the aggregation in each round, which is not real in either cross-device or cross-silo scenarios. It is necessary to also show the performance under different client participation rates.

2. Different number of clients in different datasets.

In CIFAR10/CIFAR100/iNaturalist-2017, the client number are set to be different(100/50/20) without explanation. More justification on what kind of setting(i.e. cross-silo or cross device) are these experiments trying to mimic on is needed.

3. Other KD related baselines.

Since FedSCD borrows the idea of knowledge distillation, the author should also consider to include baselines also incorporate KD such as  FedKD[2] and FedMD[3]

4. The different non-iid setting on different dataset.

In Table 1, experiment with CIFAR-10, the author shows the result of Dirichlet distribution with alpha = 0.1, 0.5, 1. However, in Table 2, experiment with CIFAR-100, only alpha = 0.1 and 1 are shown. In Table 3, experiment with iNaturalist-2017, only alpha = 0.1 is shown. I would like to see the authors to also make the heterogeneous settings in CIFAR-100 and iNaturalist-2017 as diverse as the setting in CIFAR-10.

---


[Theoretical Analysis]
1. Convergence Analysis of spectrum distillation in FL.

Spectrum analysis itself has a strong theoretical cornerstone, and at the same time, the methodology of this paper is a significant alteration of traditional FL practice. I would look forward to seeing the authors do some analysis or discussion of the theoretical aspects of the effectiveness of FedSCD, such as the rate of convergence.

2. More intuitive explanation of the accuracy drop in table 5.

After removing the PM-SCD in ablation study, although the GM-SCD is applied, the overall performance drops compared to naive setting where both PM-SCD and GM-SCD are removed. It will be nice to provide some intuitive explanation of such accuracy drop.

---
Reference:
1. SWIFT: RAPID DECENTRALIZED FEDERATED LEARNING VIA WAIT-FREE MODEL COMMUNICATION

2. FedKD: Communication Efficient Federated Learning via Knowledge Distillation

3. FedMD: Heterogenous federated learning via model distillation


========================Score Modification=======================

The author's supplementary experiments and subsequent responses after the rebuttal phase have addressed a number of my concerns. Consequently, I am inclined to elevate my score to a 5. Nonetheless, I still would like to urge the author the author to fully implement the pledged amendments and conduct comprehensive additional experiments for the next versions. Also, I recommend giving further attention to the other issues that I raised yet remain unaddressed such as the theoretical analysis.

**Questions:**

In Table 5, the method without both SCD-PM and SCD-GM share same GM acc with method without SCD-GM and same PM without SCD-PM. But intuitively, since the training scheme is different, the acc should be different. Can you explain this?
Please also refer to the Weakness section.

---

> ### Author Rebuttal · Authors · 2023-08-09
>
> We thank the reviewer for the detailed comments and suggestions.
> > Why spectral co-distillation is not counter-intuitive
>
> In existing PFL works, a common starting point is that the generic model (GM) and personalized models (PMs) share the same low-level features, while the high-level features are assumed to be more client-specific. In such works, the low/high-level features are associated to different parts of the model. It is natural to only use low-level features for GM training, since after all the low-level features are shared, while high-level features correspond to personalization. In our work, we adopt the same starting point but instead relate low/high-level features with low/high-frequency components of the spectrum, which is well-justified by existing theory on spectral bias. Thus our work is consistent with the general PFL intuition.
>
> Although high-frequency components correspond to personalization, the averaged high-frequency components would still play the role of regularization. This is crucial for minimizing the risk of overfitting PMs with limited data.
>
> We also performed an ablation study to understand the effect of partial (low-frequency only, high-frequency only) versus full spectrum for SCD-PM, on CIFAR-10 with $\alpha=1$.
>
> | Setup | Low only | High only | Full  |
> | ----- | -------- | :-------- | ----- |
> | Acc   | 90.23    | 91.05     | 92.25 |
>
> These results show the efficacy of our SCD-PM design.
>
> > Wait-free protocol
>
> As pointed out, the wait-free protocol is not our main focus. However, we still choose to include it, since it is a natural and easy extension that uses the two distillation stages in our co-distillation framework. As highlighted by Reviewer rUNY as a strength, our protocol is a general method that can be adopted by other PFL methods. We thank the reviewer for highlighting SWIFT. In Sec. 3.4 of our paper, we have cited SWIFT.
>
> >Describing Fourier spectrum for unfamiliar readers
>
> We thank the reviewer for the suggestion. We plan to build on lines 42-48 and give further elaboration in our revision.
>
> > Partial client participation
>
> To address the reviewer's concern, we give further results (see below) for partial participation with two typical participation rates ($c$=0.1 and 0.2) on CIFAR-10 with $\alpha=1$.
>
> Methods   $c$=0.1   $c$=0.2
> Setup       GM   PM   GM   PM
> FedAvg    83.16   78.25   83.84   79.03
> FedProx   82.97   78.23   83.75   79.44
> FedRoD   84.22    89.31   84.90   89.68
> Ours        84.35   89.56   85.11    90.37
>
> These results clearly show that our method is applicable for partial participation, and even outperforms our baselines. We shall give an analysis in our revision.
>
> > Clarification on experiment setup
>
> We chose different numbers of clients $N$ on different datasets to show that our method generalizes well on diverse system scales, including both cross-device ($N=100$) and cross-silo ($N=20$) FL settings.
>
>
> > Other KD-related baselines
>
> We agree that incorporating KD-related baselines would be good. However, for both suggested methods (FedKD, FedMD), we believe they are not suitable.
>
> FedKD is evaluated only on NLP tasks, and their code needs major modifications before it can be applied to the CV tasks for our experiments. FedMD requires public data, thus it is not a fair choice for other baselines without this requirement.
> To address the concern, we instead choose FedGen [1] as an additional KD-related baseline.
>
> A comparison of CIFAR-10 is given below. We shall give an analysis in our revision.
> |        | $\alpha=1$ |       | $0.5$ |       |
> | :----- | ---------- | ----- | ------------ | ----- |
> |        | GM         | PM    | GM           | PM    |
> | FedGen | 86.24      | 85.79 | 80.86        | 79.14 |
> | Ours   | 86.37      | 92.25 | 81.27        | 86.59 |
>
> [1] Data-Free Knowledge Distillation for Heterogeneous Federated Learning
>
> > Diverse non-IID settings
>
> To enhance the demonstration of our outperformance in diverse non-IID settings, we performed further experiments on FEMNIST. Overall, this means our method outperforms the baselines over **4** datasets, 2 of which involve real-world settings. Due to limited time, we are unable to perform cost-intensive experiments on iNaturalist as requested. For results on FEMNIST, see table in response to Reviewer rUNY.
>
> > Theoretical analysis
>
> We agree that further theoretical analysis would benefit our work. We are encouraged that the reviewer thinks our approach is a significant alteration of FL practice. Since our experiments show significant improvement over baselines adopting the traditional PFL paradigm, we believe our work would attract further attention, not just for the development of new PFL methods, but also for further theoretical interest in the role of spectral information in PFL.
>
> > Clarification for ablation study
>
> In the setup where both SCD-PM and SCD-GM are removed (Case i), the GM training is identical to FedAvg. In the case of removing only SCD-PM  while keeping SCD-GM (Case ii), each PM would be trained locally without any knowledge distilled from the GM. This is akin to the client training its model by itself, separately from the server. Naturally, the PM performance would be drastically lower.
>
> As SCD-GM is kept in Case ii, where the GM is the student and the PM is the teacher, since the PM's performance is drastically lower, we would expect a drop in the GM's performance. Informally, the model would be worse off with the distillation of bad knowledge, than without distillation.
>
> We shall give a detailed clarification in our revision.
>
> > Clarification for ablation study II
>
> It is not true that the training schemes are different.
>
> Without SCD-PM, each PM will be trained only by itself, separately from the server. Thus, with or without SCD-GM, the PM's training is exactly the same. So we expect the same accuracy. Analogously, without SCD-GM, the GM's training is identical to FedAvg. Thus, with or without SCD-PM, we expect the same accuracy for GM.

---

> > ### Comment · Reviewer_8564 · 2023-08-21
> >
> > Thanks to the author for making additional experiments and responses, its addressed some of my concerns. I am willing to increase my score to 5.

---

> > > ### Author Response · Authors · 2023-08-21
> > >
> > > We thank the reviewer very much for the very favorable increase in score. We greatly appreciate the reviewer's valuable comments/suggestions that would undoubtedly help improve the paper. We will include all additional ablation studies and all promised comprehensive experiment results in our revision. As for theoretical analysis, please note that we have some elaboration on a convergence analysis in our response to Reviewer jcpB. We hope this elaboration, which we plan to include in the revised supplementary material, would address the reviewer's remaining concern.

---

### Comment · Area_Chair_himT · 2023-08-21
**to reviewers**

Dear reviewers,

Thank you for your service in reviewing for NeurIPS this year!

Please carefully read the authors' rebuttal and respond to them by the followings:
1) Acknowledge that you have read the rebuttal
2) Indicate whether and how you plan to change your score (if necessary)
3) Ask the authors follow-up questions (if any)

Thanks again!

Sincerely,
AC

---

### Decision · Program_Chairs · 2023-09-21

**Decision:**

Accept (poster)

**Comment:**

In this submission, the authors propose a co-distillation method to better deal the relations between global model and personalized models. Compared with existing pFL methods, this is a new perspective to examine such relations. The presented evaluations also demonstrate its effectiveness and great potentials. To this end, I recommend accepting this submission.

For the final decision, I lower the weight of reviewer FUrR as his comments are short and less meaningful. But all the other reviewers are pretty helpful. Hope the authors find the discussion with reviewers useful and make this submission a better one.